# Prevalence of major depressive disorder and its determinants among young married women and unmarried girls: Findings from the second round of UDAYA survey

**Shromona Dhara** [1]*, **Joyeeta Thakur**[2], **Neelanjana Pandey**[3], **Arupendra Mozumdar**[4], **Subho Roy**[5]

1 Department of Anthropology, University of Calcutta, Kolkata, West Bengal, India, 2 Indian Council of Medical Research, Hyderabad, India, 3 Population Council Consulting Pvt. Ltd., New Delhi, India, 4 FHM Engage, New Delhi, India, 5 Biological Anthropology Unit, Indian Statistical Institute, Kolkata, India

* shromona.dhara@gmail.com

**Data Availability Statement:** This study was performed by analyzing the secondary data of

## Abstract

### Introduction

Depression is a prevalent and debilitating mental illness affecting young women worldwide. This study aimed to identify psychosocial determinants of major depressive disorder (MDD) among young women in Bihar and Uttar Pradesh, India.

### Methods

Data from "Understanding the Lives of Adolescents and Young Adults" (UDAYA) study (2018-19) for young women aged 12-23 years, both married and unmarried was used for this paper. MDD was evaluated using the Patient Health Questionnaire PHQ-9 with a cut-off score of $\leq 10$. The determinants of MDD were identified through multilevel binary logistic regression analysis.

### Results

The prevalence of MDD was 13.6% (95% CL 12.2-15.2) and 5.1% (95% CL 4.2-6.1) for young married women and unmarried girls, respectively. Among the young married women, community-level variables like dowry-related humiliation (1.74, 95% CI 1.15-2.64), and sexual assaults (2.15, 95% CI 1.24-3.73) were significantly associated with MDD. For unmarried girls, reporting of family violence $_{<10\% \text{ of participants}}$ (0.45, 95% CI 0.24-0.85), family violence $_{(\geq 10\% \text{ of participants}) \%}$ (0.35 95% CI 0.19-0.68) and interpartner violence $_{(>25\% \text{ of participants})}$ (0.42; 95% CI 0.23-0.74) remain significant predictors of MDD. At individual level, for both the groups, age, participation in decision making (on education), social capital (currently attending school/educational course and number of friends), self-efficacy, telephonic harassment, and physical activity were associated with MDD. Wealth index, job seeking, participation in decision making (on health-seeking), parental interactions and physical abuse (for unmarried

UDAYA Study (Understanding the Lives of Adolescents and Young Adults), conducted by the Population Council. The data are available in public domain and can be accessed on request to the Council. The link to access the UDAYA data is given below https://dataverse.harvard.edu/dataset.xhtml?persistentId=doi:10.7910/DVN/ZJPKW5 DOI: 10.31899/pgy8.1045.

**Funding:** This study received no funding from any external source.

**Competing interests:** he authors have declared that no competing interests exist.

girls only) and education, reported last sexual intercourse, pressure from the in-laws' to conceive (for young married women only) were associated with MDD.

## Conclusions

For young married women, community level targeted interventions should focus on the social ecology to foster a sense of safe community environment. For unmarried girls, additionally, interventions should aim to optimize their family environment for effective mental health outcomes.

## Introduction

Depression is a widespread mental health condition that has become the most common cause of disability in the world. On a global scale, an estimated 280 million individuals encounter depression annually [1]; and from 2000 to 2020, approximately 34% of adolescents of the world self-reported experiencing depressive symptoms [2]. According to WHO (2011), in India, 20% of young individuals may encounter various mental health challenges, including depression, mood swings, substance abuse, suicidal tendencies, eating disorders [3]. Social stigma and unmet need for treatment of mental health condition are going to be the leading cause of global disease burden by 2030 [4]. A poor mental state could make a person suffer profoundly either in terms of relation with family and friends, performance at work, substance abuse, non-suicidal self-injury and in the worst scenario, suicide, or it may lead to other health issues like cardiovascular morbidity [1, 5–7].

In India, despite the implementation of the National Mental Health Program almost four decades ago, the prevalence of mental disorders continues to grow [8]. With the highest suicide rate in south-east Asia [9], this increase is not only because of social exclusion, oppression, gender inequities, pre-existing conditions such as chronic illnesses and family history but also attributed to poor implementation existing of mental health services [10–12]. The National Mental Health Survey in 2015-16 [13] reported that almost 85% of such cases remained undiagnosed or do not receive any treatment despite their severity of suffering.

Individuals' mental health and well-being are functioned by an integrated effect of the personal, physical, and shared social environment [14]. Multiple bio-psycho-social factors like poverty, unemployment, financial strain, gender disparity, strained familial relationship characterized by parental fights, domestic violence, or physical abuse by parents, inordinate screen time, internet and social media addiction and associated phubbing behavior coupled with less physical activities, and stressful life events increase the risk of depressive behavior and its long persistence [10, 15–28]. Among the school and college-going kids, depressive symptoms and anxiety possess a bilateral causal relationship with decreased physical activity and higher screen time [20, 22, 29]. Individuals, hailing from lower social strata in terms of caste or religious minority group, mediated through lack of social support, deprivation of educational or economic resources, and stressful life events, potentially impact mental health [17, 19] On the contrary, there are some factors that act as protective buffers by developing personal resilience and prevent depressive disorders [30]. Individuals' personals traits such as, the dimensions of cohesion, expressiveness, and independence within the family environment is negatively correlated with experiencing depression [31]. Possession of social capital entails a person having resources from a viable network of social ties, recognition, and mutual acquaintance [32, 33]. A conscious construction of such sociability, mediated by active participation in decision-

making, self-efficacy and supportive interaction with parents act as a cognitive mechanism to control stress and conceivably reduce the risk of depression by creating mutual trust and emotional support, especially during the transition into adulthood [34–39].

Studies reported that the Major Depressive Disorder (MDD) is more commonly experienced by emerging adults than in adults or children owing to rapid social, emotional, and cognitive development and key life transitions [40, 41]. Women have a significantly higher prevalence of major depression owing to the differences in physical strength, personality traits, and diet [21, 42, 43]. During puberty and pregnancy, depression is twice in women than in men, which may be attributed to variations in ovarian hormones, especially decreased estrogen levels [44, 45]. At the same time, the set of roles and responsibilities that society assigns to a married woman create a distinct social-ecological context. Getting married at an early age and thereafter leaving the desired education or career behind, carrying the burden of fulfilling familial expectations, staying away of husband from home for work and giving birth to a child, or instead receiving snide remarks from the in-laws expose a woman to the catastrophe of mental stress [46–52].

Living in a community that embraces a shared social life and collective identity, women seek a secure space, ensured with minimized gender-based crime, dowry harassment, domestic or spousal violence, and endorsement and recognition of women's autonomy [53]. Devoid of such attributes, the community environment transmutes into an extremely intimidating realm and propels women into lifelong emotional distress of traumatic experiences [54–56].

Data from the 2020 India State Level Disease Burden Initiative revealed a concerning profile of mental health in India. One in seven individuals nationwide experiences a mental disorder, ranging from mild to severe. Depressive and anxiety disorders, typically emerging during adulthood, were the most common mental health conditions. The study also found a notable disparity: childhood and adolescent-onset mental disorders were more prevalent in less developed northern states like Bihar, and Uttar Pradesh, compared to their prevalence in more developed southern states [12]. This study adopted Bronfenbrenner's Ecological Systems Theory (EST) integrated with subtle contextual modifications. EST is a widely used nested theoretical model which explains how an individual's health and mental health is shaped by the interaction of five influential levels of ecological systems: micro-, exo-, meso-, macro-, and chrono-systems [57]. Therefore, the psychosocial epidemiology of depressive disorder entails disharmony within the resources innate to their social-ecological levels [58] i.e the complex interplay between an individual and the multiple layers of environmental system [59]. There are studies carried out to understand the proximal impact of this ecological model and its components in examining depressive disorders across the globe [60–63].

On the basis of the EST model, we hypothesized that MDD is a result of dynamic and active engagement with personal, household, and community-level environments (Fig 1). The microsystem includes individual-centric factors (e.g., sex, age, health, and the intimate or immediate world like parents, siblings, spouse, or friends); the exo-system represents the organizational behavior of coexisting within several social structures and the macro system incorporates the community, the larger social context and prevailing norms to which individuals orient themselves. Disruption in communication between and within these systems reflects social disorientation or anomie and resulted in depression.

The survey data from "Understanding the Lives of Adolescents and Young Adults" (UDAYA) study, offers an opportunity to understand the factors assigned to different spatial and ecological levels that are likely to determine MDD among adolescents and young adults in two Indian states namely Bihar and Uttar Pradesh. Since the survey recruited one individual from a household, we considered the household information as individual-level data and framed it into two ecological levels. A schematic diagram illustrating the multilevel structure of the data is given in the Fig 2.

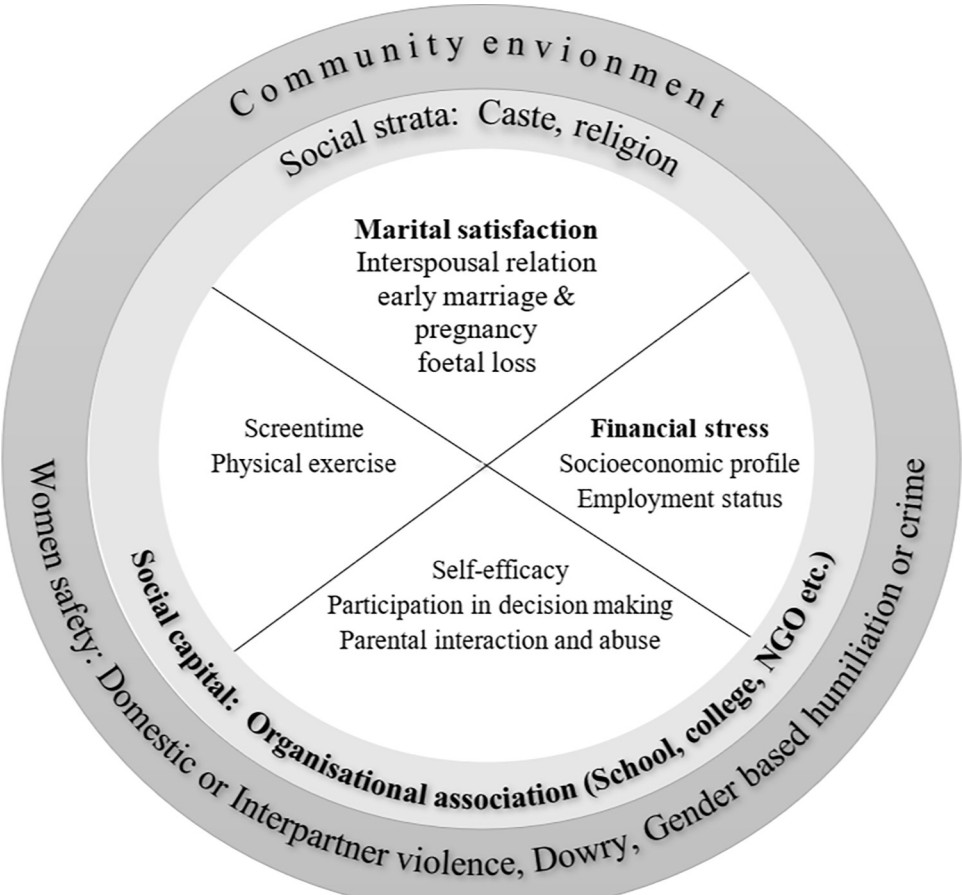

**Fig 1. Psychosocial determinants of depression interplaying at different social-ecological levels.**

Against this backdrop we hypothesized that, MDD among the married women and unmarried girls living in the states of Bihar and Uttar Pradesh is influenced by a range of psychosocial determinants at both individual and community levels.

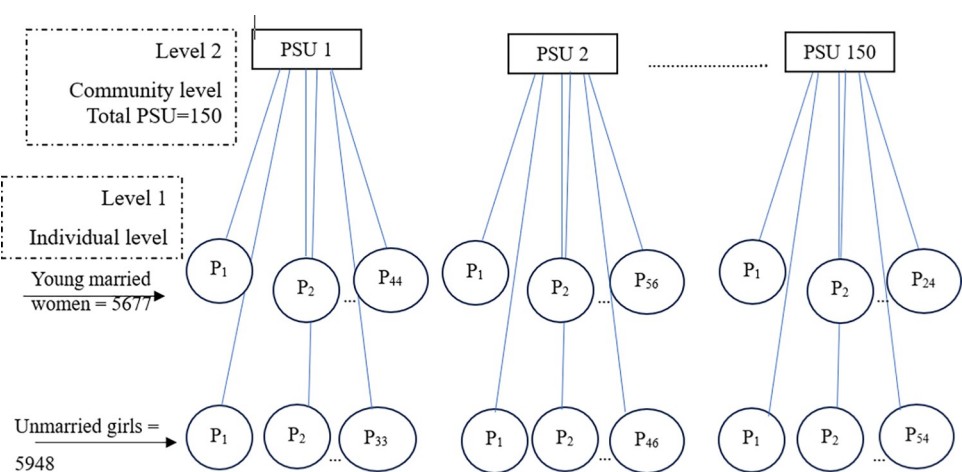

**Fig 2. A schematic diagram of the nested multilevel structure of the data.**

## Materials and methods

### Data source

This study used longitudinal survey data from the "Understanding the Lives of Adolescents and Young Adults (UDAYA)" that was conducted in two Indian states namely Bihar and Uttar Pradesh during 2015–16 and its follow-up survey was conducted almost after three years, during 2018–19 by the Population Council. In this paper, we analyzed data from the adolescents and young adults who were interviewed successfully in the second round UDAYA-2 survey. The data set was retrieved for analysis on May 24, 2021.

The UDAYA survey adopted a multi-stage systematic sampling design to provide state-level estimates covering samples from both urban and rural areas. The survey was conducted in 150 select primary sampling units (PSU)—75 from urban and 75 from rural areas. The details of the study design and sampling strategy of the UDAYA are available in the study report [64, 65].

### Study participants

The UDAYA, 2018–19 survey re-interviewed 16,292 boys and girls aged 12–23 years. The following steps (Fig 3) were taken to obtain the analytical sample for the present study. We excluded data of telephonic interviews (n=220) to maintain consistency among the participants by mode of data collection. A set of 119 females were excluded, who were ever married but currently not in wedlock, i.e., being married but no *gauna*, separated, deserted, divorced, or widow. The category married but 'no *gauna*', refers to those young married women who are yet to leave their natal place or start cohabiting with their husbands, a prevalent cultural practice in the states of Bihar, Uttar Pradesh. The final analytical sample consists of data from 5,677 young married women and 5,948 unmarried girls. At the time of survey, the young girls and married women aged between 12-23 years. Additionally, we also used the data from 4,128 unmarried boys aged 12-23, who participated in the same round of survey, to estimate some independent variables, assuming their potential impact on mental health of the young married women and unmarried girls.

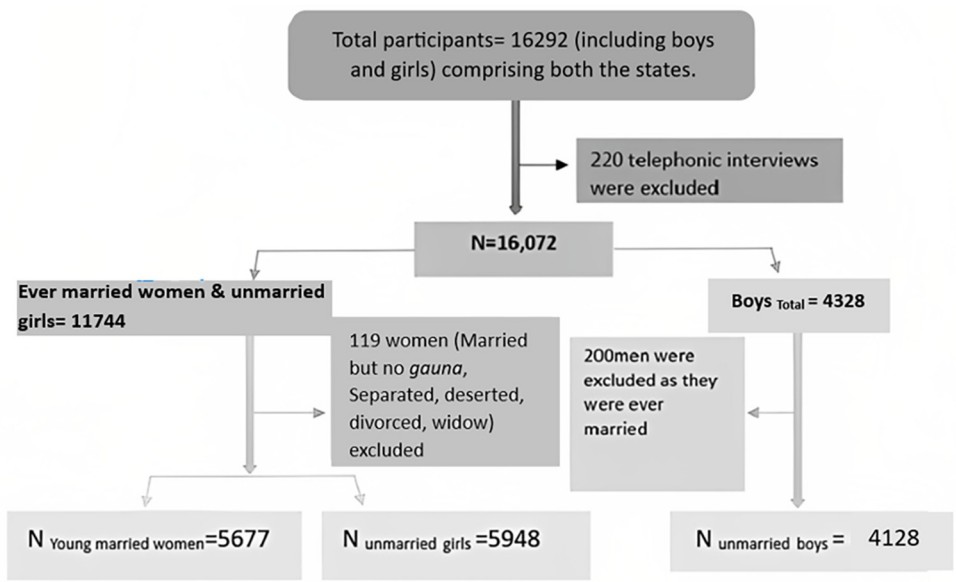

**Fig 3. Inclusion and exclusion criteria of the study population.**

## Variable description

**Dependent variable.** The dependent variable of this study was the level of depression, assessed through the patient health questionnaire (PHQ-9 score). Participants were asked about the frequency of experiencing nine depressive symptomatic behaviors experienced during the last 15 days preceding the survey using a 5-point Likert scale [66–68]. The list of symptoms, recoding of the responses, and assessment of a composite score for the outcome variable have been provided in S1 Table. This score ranged between 0 and 27. The degrees of depression were classified as no/mild (scores between 0 and 9), moderate (scores between 10 and 14), moderate to severe (scores between 15 and 19), and severe (scores 20 and above). We used a cut-off ≥0 to identify the individuals with MDD. This cut-off demonstrated an effective screening for major depression with 88% sensitivity and 88% specificity in earlier studies [66]. In this analysis, we considered the outcome variable, i.e., 'had MDD' as binary with the same cut-off (participants with ≥10 score was coded as '1' and those with < 10 as 'no and/mild depression).

**Independent variables.** *Independent variables at the individual level.* We considered a battery of independent variables that are expected to be associated with the dependent variable at the individual level. These variables include demographic and socioeconomic factors (age, social group, religion, educational status, employment status, and household wealth), social-capital-related variables (current schooling, and the number of friends), agency-related variables (financial control, decision-making, and self-efficacy), the experience of domestic violence, and other variables like the exposure of screen-time, the experience of pregnancy loss, and experience of live births. A list of these independent variables along with their respective categorizations is mentioned in S1 1 Table of S1 File.

We recoded some variables for ease of analysis. For example, we coded the missing values for 'financial control over the money earned in last year' as 'no', considering those participants as they did not have any source of income in the year preceding the survey. Experiencing harassment over the telephone and/or receiving improper pictures or texts, either through mobile or the internet, were merged and categorized as 'yes'; the rest who did not experience any such were coded as 'no'. All the decision-making variables were categorized as 'yes', if the participant took the decision alone or jointly with others; otherwise categorized as 'no.' Decision-making on how much education one should attain was coded as 'no' for those participants who were not currently attending any school. Parental interaction was ascertained if the participants had discussed certain aspects of their life with both or either of the parents. Unmarried girls who did not discuss or interact with any of their parents were coded as 'no'. A detailed description of the questions on self-efficacy along with their respective coding is provided in S2 Table.

*Independent variables at the community level.* The primary sampling units (PSUs) are the clusters of households that share a common social-ecology. The hierarchical data structure of UDAYA allows individuals to be nested within the PSUs. Therefore, for this analysis, we considered the PSUs as proxy for the communities and generated community-level variables by aggregating individual-level data at the PSU level.

Two sets of explanatory variables were generated at the community level, one for married and another for unmarried participants. Selection of community-level variables for the multi-level modelling was guided by the following criteria—first, we focused on characteristics of the communities that could not be measured for all participants at the individual level. Second, these characteristics should reflect aspects of the community that would likely to influence mental health outcomes for all residents. Aggregating these variables to the PSU level, allowed us to examine their effects at the community level. Some of these variables were common to

both married and unmarried participants (experienced any sexual assaults, dowry-related humiliation, men's gendered attitude, and intimate partner violence (IPV). However, for young married women some additional variables such as, family unrest and sibling inequality were also considered as community level independent variables. The list of variables, aggregated at community level, along with their respective categorizations is presented in S2 Table.

To assess the prevalence of sexual assault at the community-level, a composite variable was created by merging three items from the girls' questionnaire in UDAYA. These items asked whether the woman had experienced eve-teasing, molestation, rape, or attempted rape by a stranger in the past three years. The percentage of participants who reported at least one of these kinds of sexual assault was calculated at the community level. Similarly, dowry-related humiliation was assessed by asking whether the woman had to pay a dowry or have been humiliated by her in-laws for 'insufficient' dowry. The percentage of young married women who responded 'yes' to any of these two questions (or both) was estimated. Different types of IPV were assessed from the reported experience of the participants for seven variables related to physical violence, one variable each for emotional and sexual violence in the last year preceding the date of the survey. The participants were coded '1' if reported each of these nine IPV experiences, otherwise coded as '0'. Adding the codes we calculated a composite score of IPV which could possibly range from 0 to 9 where '0' indicates no experience of IPV and 9 for experiencing all forms of IPV. A detailed description of the types of violence, their coding, and assessment of the composite score on IPV have been enlisted in S3 Table. The participants were categorized into high IPV and low IPV categories, based on the median value of the composite score. Finally, the percentage of young married women who experienced high IPV was estimated at the community level. Likewise, the data on gender attitudes were collected, by asking attitudes of the boys on nine statements (S4 Table). Boys who responded 'not sure' about any question and/or, do not possess a gendered perception towards women for that question were grouped into 'no' (coded as '0'). The median of the composite score (based on the responses to nine questions) was used as a cut-off value to delineate boys who had more and less gendered opinions toward women. The percentage of boys with more gendered opinions towards women was estimated at the community level. Among the unmarried girls, perceived sibling inequality was assessed among those who have at least one brother/sister/cousin residing with them. Since some unmarried girls do not have any sibling, the variable was estimated at the community level instead of at the individual level. Girls who were receiving at least one out of five types of inequal treatments from their parents were coded as '1' (inequality perceived), otherwise '0'. A list of perceived unequal treatment is listed in S5 Table. For the young married women, we have excluded the aggregated variables on perceived sibling inequality and family unrest.

## Statistical analyses

We used descriptive statistics separately for married and unmarried groups. For the estimated variables, aggregated at the community level, the categorization of the PSUs was done after examining the distribution of the participants. We used t-test and chi-square tests to compare the depression levels between the two groups of participants. The UDAYA study provides the sampling weights with the dataset. Appropriate weights were applied for each estimation to account for the sampling design of the study and to ensure that the estimates are representative of the study population. Some of the continuous variables (such as the number of friends, hours of physical activity and screentime, domestic violence, and men's gender attitude) were converted into binary variables based on weighted median values independently for married and unmarried participants. The multicollinearity among the independent variables was measured through the variance inflation factor (VIF > 10) and its respective tolerance (<0.1).

We applied two-level mixed-effects (random and fixed effects) multilevel logistic regression analyses, specifically with random intercepts, for married and unmarried groups separately to examine the effects of individual and community-level explanatory variables on the level of depression. The random intercept captures this variation across communities without any independent variable, while fixed effects estimate the effects of individual-level variables on depression, holding community-level factors constant. We applied multilevel modelling because some of the independent variables were not directly measured at the individual level, yet they reflect characteristics of the social ecology that affects the variables determining the depression at the individual level. The nested structure of the UDAYA data allowed the analyses using multilevel regression models to examine the effects of various contextual variables operating at the community level apart from determinants at the individual level [69–71]. We hypothesized that participants from the same PSU may have similar exposure to community norms and societal expectations; thereby, the participants will have differential exposure across PSUs.

The mixed-effects multilevel logistic regression function can be written as:

$$\log\left(\frac{\pi ij}{1 - \pi ij}\right) = \beta_0 + \beta_1 X_{1ij} + \ldots + \beta_n X_{nij} + \mu_{0j} + e_{ij}$$

Where, $\pi_{ij}$ is the probability of having moderate to severe depression, $\beta_0$ is the log odds of the intercept, $\beta_1 \ldots \beta_n$ are the coefficients for the respective independent variables at the individual and community level, $X_{1ij} \ldots X_{nij}$ are the independent variables—$\mu_{0j}$ and $e_{ij}$ are random errors at the community and individual levels respectively.

We calculated the intraclass correlation coefficient (ICC) and proportional change in variance (PCV) for the respective models which provide a random effect estimation of the variance proportion in the level of depression across the PSUs. ICC measures the proportion of variance in the outcome variable that can be attributed to group membership in a multilevel model. The ICC is defined as the variance between clusters ($\sigma^2$ between) divided by the total variance, the summation of the variance between and within clusters ($\frac{\pi}{3}$). Smaller the variance within the PSUs greater the ICC would be. The ICC is calculated as:

$$\text{ICC} = \frac{\sigma^2{}_{\text{between}}}{\sigma^2{}_{\text{between}} + \frac{\pi^2}{3}}$$

PCV which estimates the percentage of variability in the odds of the outcome variable explained by the successive models with respect to the empty model [72] is calculated as:

$$PCV = \frac{V_e - V_{mi}}{V_e}$$

[$V_e$ = variance of the empty or 'null' model, and $V_{mi}$ = variance of the individual model with more terms.]

ICC provides insight into the strength of clustering, while PCV highlights the explanatory power of community-level factors in the model [72, 73].

The log-likelihood (-2LL) ratio test was used to compare the models and determine the best fit. The likelihood ratio test compares the 2 log-likelihood of the model with a random intercept and the 2 log-likelihood of the preceding model, following a Chi-square distribution [74, 75]. The Akaike Information Criterion (AIC) was used to compare the models and the final model with the lowest AIC has been considered as best fit [76, 77].

The Statistical Package for Social Sciences version 26.0 (SPSS Inc., Chicago IL, USA) was used for the analyses. The fixed effect sizes of individual and community-level variables were

expressed by the adjusted odds ratios (AOR) with 95% confidence intervals [78]. A *p*-value of < 0.05 was considered statistically significant.

Three regression models were calculated. The first model was the null model or Model 0 which was fitted without any explanatory variables. It estimated the total variance of the severity of depression and broke it down into two components: variance within (residual) and between groups (variance: intercept). Based on the variance of the intercepts and the remaining error variance, ICC and PCV were estimated for each of the subsequent models, i.e., Models 1, 2, and 3.

Model 1 was constructed with individual-level independent variables. In Model 2, only community-level independent variables were added. In Model 3, the full-model, we included both individual and community-level independent variables. All three models were constructed with two distinct sets of exploratory variables (as described in S1 1 and S1 2 Tables of S1 File) separately for married and unmarried groups.

## Ethical considerations

This paper is an outcome of secondary analyses of the data available in the public domain, devoid of any access of the authors to the personal information that could identify individual participants (all data were fully anonymized before it was accessed) during or after data collection. The Population Council Institutional Review Board provided ethical approval for the study. Adolescents provided individual written consent to participate in the study, along with a parent/guardian for unmarried adolescents younger than 18. All consent procedures for the primary data collection activities were reviewed and approved by the Institutional Review Board at the Population Council (protocol number 698).

## Results

### Characteristics of the young married women

Among the young married women, about 2% were aged below 18 years (i.e., below the legal age at marriage in India). About 82% of married women were Hindus, 54% belonged to other backward classes and 40% were from the lower socio-economic strata with respect to the state-wise wealth index. About 37% of the young married women completed schooling up to 8th standard, 25% completed education up to 12th standard and beyond; whereas only 14% of young married women had done any paid work in the last year and 15% were looking for a job (Table 1).

At the time of the survey, only 5% of young married women were pursuing school education or any coursework, around 9% had been a member of any youth/women's group in the last three years and approximately two third of the young married women had less than four friends. About 13% of young married women could take decisions on how to spend their earnings; whereas more than half of the married women took part in deciding to which standard they should study and whether they should work. Most of the young married women (43%) had the 'middle' level of self-efficacy which scored between 16 and 20. Around 6% of the young married women experienced telephonic- or cyber-bullying in the last three years. Most of the young married women (89%) reported less than one hour of physical activity and about 53% had more than two hours of screentime in the last week.

More than half of the young married women got married between 15 and 17 years. One-third of the young married women living with husband were staying alone since in most of the cases their husbands stay at their place of job. Nearly 46% of couples reported having last intercourse in terms of 'days' ago. More than three-fourth (78%) of the young married women gave at least one live birth, 21% experienced at least one miscarriage and around 4% had at least one

**Table 1. Distribution of young married women and unmarried girls by various (individual level) background characteristics living in Bihar and Uttar Pradesh.**

| | Married (Unweighted N=5,677) | Unmarried (Unweighted N=5,948) |
|---|---|---|
| | Weighted % | Weighted % |
| *Socio-economic characteristics* | | |
| Age in completed years | | |
| 12–17 | 2.1 | 70.2 |
| 18–23 | 97.9 | 29.8 |
| Social group | | |
| Scheduled castes (SC)/tribes (ST) | 29.5 | 23.40 |
| Other backward classes (OBCs) | 57.3 | 53.40 |
| General | 13.3 | 23.20 |
| Religion | | |
| Hindu | 81.6 | 74.3 |
| Non-Hindu | 18.4 | 25.7 |
| Completed years of schooling | | |
| No schooling | 19.5 | 4.2 |
| Standard 1 to 8 | 37.2 | 47.0 |
| Standard 9 to 10 | 18.3 | 22.7 |
| Standard 11+ | 25.0 | 26.1 |
| State specific wealth index | | |
| Low | 39.7 | 31.6 |
| Middle | 31.0 | 30.7 |
| High | 29.2 | 37.6 |
| Working status | | |
| Done paid work in formal sector (in last one year) | 14.3 | 23.6 |
| Currently looking for a job | 15.4 | 11.0 |
| *Social capital* | | |
| Currently attending any school/educational course | 5.3 | 65.6 |
| Number of friends | | |
| 0–3 | 62.1 | 47.8 |
| 4 or more | 37.9 | 52.2 |
| Member of youth/women's/girls' group in the last three years | 8.8 | 2.5 |
| *Agency* | | |
| Took decision on how money earned the last year should be used | 12.5 | 18.4 |
| Took decision on how much education should have/ till which standard | 50.3 | 59.2 |
| Took decision about who would be friends | 96.1 | 96.7 |
| Took decision about making major household purchases | 44.9 | 28.3 |
| Took decision about whether to work or stay at home | 53.9 | 53.9 |
| Took decision about seeking treatment if feeling sick | 47.5 | 38.9 |
| Self-efficacy scores | | |
| 5–15 | 34.1 | 20.6 |
| 16–20 | 43.3 | 45.2 |
| 21-24 | 22.6 | 34.2 |
| *Experience of harassment* | | |

*(Continued)*

**Table 1.** (Continued)

| | Married (Unweighted N=5,677) | Unmarried (Unweighted N=5,948) |
|---|---|---|
| | Weighted % | Weighted % |
| Experienced telephonic or online harassment (in last three years) | 5.5 | 6.1 |
| *Physical activity* | | |
| Duration of heavy work in last week | | |
| Less than an hour or no work | 88.3 | 83.1 |
| More than one hour | 11.7 | 16.9 |
| Duration of screentime in last week | | |
| 0 to 2 hours | 47.1 | 36.9 |
| 3 to 15 hours | 45.0 | 51 |
| more than 15 hours | 7.9 | 12.1 |
| *Interaction with parents in last year (only for unmarried)* | | |
| Discussed about school performance | NA | 58.7 |
| Discussed about friendship during | NA | 65.2 |
| Discussed about menstruation | NA | 62.2 |
| Discussed about being teased | NA | 17.9 |
| Discussed about how pregnancy occurs | NA | 1.8 |
| *Physical abuse by parents* | | |
| Physically hurt by parents in the last one year | NA | 11.5 |
| *Reproductive history (only for married)* | | |
| Age at marriage (in years) | | |
| 20-22 | 5.6 | NA |
| 18-19 | 25.8 | NA |
| 15-17 | 56.1 | NA |
| 7 to 14 | 12.4 | NA |
| Currently living with husband | | |
| Couple lives together | 68.0 | NA |
| Couple stays apart | 32.0 | NA |
| Reported last sexual intercourse | | |
| Days ago | 45.7 | NA |
| Weeks ago | 18.6 | NA |
| Months ago | 33.9 | NA |
| Years ago | 1.9 | NA |
| Ever given birth to a live child | 77.6 | |
| Had experienced miscarriage/stillbirth | 21.2 | NA |
| Had experienced induced abortion | 3.9 | NA |
| In-laws pressurized to conceive immediately or would call infecund | 20.0 | NA |

induced abortion. One-fifth of all the young married women were pressurized for conception or had a fear to be called barren by their in-laws in case remain childless.

## Characteristics of the unmarried girls

The unmarried girls, who participated in the survey, were mostly between 12 and 17 years (70%); more than 80% were Hindus and the majority belonged to other backward classes

(53%), followed by those who belonged to higher socio-economic groups (38%). Only 4% of the unmarried girls did not have any schooling and most of them (47%) had schooling till 8<sup>th</sup> standard; 26% attended schooling up to 12<sup>th</sup> standard. One-fourth of them were in a paid job in the last year and 11% were looking for a job. Almost two-thirds (66%) of the unmarried girls were in school or were pursuing any educational course. Over half of them had four or more friends and 3% had been a member of any youth/girls group in the last three years.

More than 18% of the unmarried girls had reported having control over their earnings; whereas almost 60% of them could participate in deciding to which standard they should study, 54% could decide whether they should work, 28% and 39% unmarried girls took part in deciding about their household purchase and health seeking respectively. Most of the unmarried girls (45%) had a self-efficacy score between 16 and 20.

More than 17% of the unmarried girls have done more than an hour of heavy work in the last week, preceding the date of survey, and more than half had three to 15 hours of screentime. Almost two-thirds of the unmarried girls interacted with their parents regarding friendship and menstruation. Nearly 60% discussed their school performance and 17.9% interacted about being teased. Only 2% of unmarried girls reported that they had an interaction with their parents on how pregnancy occurs.

## Characteristics of the PSUs by the community level variables

In about 45% of PSUs, 25%–50% of study participants (married and unmarried inclusive) living in those PSUs, experienced any form of sexual assault. In 43% of PSUs, more than 35% of participants reported dowry-related humiliation and in 45% of PSUs, more than 25% of the participants reported inter-partner violence in the last year. In 17% of PSUs, the participants reported no family violence, i.e., they did not experience intra-parental violence (father beating their mother). In 32% of PSUs more than 30% of the participants reported any kind of sibling inequality. In about 29% of PSUs, more than 45% of the boys exhibit gendered attitudes towards women (Table 2).

## MDD among young married study participants

About 14% of young married women had MDD (Table 3). Among the older group, i.e., 18–23 years, showed a higher prevalence of MDD (14%) than their younger counterparts, i.e., 12-17 years (8%). Significant associations were observed in the prevalence of MDD with background characteristics of married girls, such as social group, religion, household wealth, and working status. The prevalence of major depression was higher among Hindus, those belonged to scheduled castes or tribes, and who had completed schooling up to the secondary level; whereas the prevalence was lower among young married women of high wealth index, if not working, or not looking for a job. The married women with more friends experienced less depression; however, those having a membership in a youth group in the last three years (preceding the date of survey) were more depressed than their counterparts. The married women who could decide on spending of their earnings (earned in the last year), were more depressed, however, the married women who took decisions on to which standard they should study, and had higher self-efficacy showed significantly less depression. The women who got married at an earlier age, had their last sex long back, and were pressurized to prove their fertility from the in-laws showed a higher prevalence of MDD than their counterparts.

## MDD among the unmarried girls

About 5% of unmarried girls had MDD (Table 3). The older group, i.e., 18-23 years, showed a higher prevalence of MDD (8%) than their younger counterparts, i.e., 12-17 years (4%).

**Table 2. Distribution of the primary sampling units (PSUs) by community level variables in Bihar and Uttar Pradesh, (Number of PSUs= 150).**

| Estimates that were calculated using data from both young married women and unmarried girls | % of PSUs |
|---|---|
| Experienced eve teasing, molestation, attempted rape, or rape in last three years | |
| Reported by less than 25% participants | 35.3 |
| Reported by 25–50% of participants | 44.7 |
| Reported by more than 50% of participants | 20.0 |
| Estimates that were calculated using data only from married girls | |
| Experienced dowry-related humiliation | |
| Reported by less than 25% participants | 42.7 |
| Reported by 25–35% of participants | 29.3 |
| Reported by more than 35% of participants | 28.0 |
| Reported inter-partner violence in the last one year | |
| Reported by less than 15% participants | 44.7 |
| Reported by 15-25% of participants | 28.7 |
| Reported by more than 25% of participants | 26.7 |
| Estimates that were calculated using data only from unmarried girls | |
| Reporting of family violence: father beaten mother | |
| Does not have parents or reported no violence | 16.7 |
| Reported by less than 10% of participants | 51.3 |
| Reported by 10% or more participants | 32.0 |
| Reporting of any kind of sibling inequality | |
| Reported by less than 0-15% participants | 27.3 |
| Reported by 16–30% of participants | 40.7 |
| Reported by more than 30% of participants | 32.0 |
| Estimates that were calculated using data only from unmarried boys | |
| Boys with gendered attitude towards women | |
| Reported by less than 25% of participants | 31.3 |
| Reported by 25 to 45% of participants | 40.0 |
| Reported by more than 45% of participants | 28.7 |

Significant associations were observed in the prevalence of MDD with background characteristics of the unmarried girls, such as social group, religion, household wealth, and working status. The prevalence of MDD was higher among non-Hindus, belonging to OBC, who did not go to school if working, or looking for a job.

The unmarried girls with more friends experienced less depression; however, those having a membership of a youth group in the last three years (preceding the date of survey) were more depressed than their counterparts. The unmarried girls who decide on spending of their earnings (earned in the last year) and those who took decisions on to which standard they should study were more depressed. The unmarried girls who took decisions on household purchases, whether to work or stay at home, seeking treatment if feeling sick and low self-efficacy showed significantly higher levels of depression. The unmarried girls who discussed their school performance with their parents were less depressed, but those who discussed friendship, being teased, menstruation, and how pregnancies occur were more depressed than their counterparts. Regardless of marital status, participants who experienced harassment over the telephone or through social media were more depressed than those who did not have such experience.

**Table 3. Bivariate statistics of young married women and unmarried girls who had experienced the major depressive disorder (MDD) by various (individual level) background characteristics living in Bihar and Uttar Pradesh.**

| Experienced major depressive disorder | Married %(CI) 13.6 (95% confidence limit 12.2-15.2) | Unmarried %(CI) 5.1 (95% confidence limit 4.2-6.1) |
|---|---|---|
| *Socio-economic characteristics* | | |
| Age in completed years | | |
| 12-17 | 8.0 | 3.8 |
| 18-23 | 13.7 | 8.1 |
| Social group [a, b] | | |
| Scheduled castes (SC)/tribes (ST) | 15.0 | 4.7 |
| Other backward classes (OBCs) | 12.7 | 5.7 |
| General | 14.6 | 4.0 |
| Religion [a, b] | | |
| Hindu | 14.1 | 4.6 |
| Non-Hindu | 11.4 | 6.6 |
| Completed years of schooling [b] | | |
| No schooling | 12.2 | 7.8 |
| Standard 1 to 8 | 13.4 | 3.8 |
| Standard 9 to 10 | 15.6 | 5.5 |
| Standard 11+ | 13.5 | 6.6 |
| State specific wealth index [a, b] | | |
| Low | 14.1 | 4.6 |
| Middle | 14.7 | 6.0 |
| High | 11.8 | 4.6 |
| Working status | | |
| Done any paid work in past one year [a, b] | | |
| Yes | 18.3 | 6.7 |
| No | 12.8 | 4.6 |
| Currently looking for a job [a, b] | | |
| Yes | 22.0 | 11.7 |
| No | 12.1 | 4.3 |
| *Social capital* | | |
| Currently attending any school/ coursework [b] | | |
| Yes | 10.2 | 4.0 |
| No | 13.8 | 7.1 |
| Number of friends [a, b] | | |
| 0–3 | 14.4 | 5.8 |
| 4 or more | 12.3 | 4.3 |
| Been a member of youth/women's/girls' group in the last three years [a, b] | | |
| Yes | 16.6 | 8.3 |
| No | 13.3 | 5.0 |
| *Agency* | | |
| Took decision on how money earned the last year should be used [a, b] | 17.6 | 5.7 |
| Did not decide herself or did not earn any money | 13.1 | 4.2 |
| Took decision on how much education should have/ till which standard [a, b] | | |
| Yes | 12.7 | 5.7 |

*(Continued)*

**Table 3.** (Continued)

| | Married %(CI) | Unmarried %(CI) |
|---|---|---|
| **Experienced major depressive disorder** | **13.6 (95% confidence limit 12.2-15.2)** | **5.1 (95% confidence limit 4.2-6.1)** |
| No | 14.6 | 4.2 |
| Took decision about who would be friends | | |
| Yes | 13.5 | 5.0 |
| No | 16.0 | 6.3 |
| Took decision about making major household purchases [b] | | |
| Yes | 13.9 | 6.0 |
| No | 13.4 | 4.7 |
| Took decision about whether to work or stay at home [b] | | |
| Yes | 14.1 | 5.6 |
| No | 13.1 | 4.5 |
| Took decision about seeking treatment if feeling sick [b] | | |
| Yes | 13.7 | 6.1 |
| No | 13.6 | 4.4 |
| Self-efficacy scores [a, b] | | |
| 0 to 15 | 15.5 | 8.5 |
| 16 to 20 | 12.1 | 4.3 |
| 21 to 24 | 13.7 | 4.1 |
| *Experience of harassment* | | |
| Experienced telephonic or online harassment (in last three years) [a, b] | | |
| Yes | 25.6 | 11.4 |
| No | 12.9 | 4.7 |
| *Physical activity* | | |
| Duration of heavy work [a, b] | | |
| No work or less than an hour | 12.6 | 5.3 |
| Any form of heavy work | 21.3 | 1.7 |
| Duration of screentime [a] | | |
| 0 to 2 hours | 14.1 | 5.6 |
| 3 to 15 hours | 12.7 | 4.1 |
| more than 15 hours | 16.7 | 5.1 |
| *Interaction with parents in last year (only for unmarried)* | | |
| Discussed about school performance [b] | | |
| Yes | NA | 4.1 |
| No | NA | 6.4 |
| Discussed about friendship [b] | | |
| Yes | NA | 5.8 |
| No | NA | 3.7 |
| Discussed about being teased [b] | | |
| Yes | NA | 9.1 |
| No | NA | 4.2 |
| Past one year, discussed about menstruation [b] | | |
| Yes | NA | 5.5 |
| No | NA | 4.3 |

(*Continued*)

**Table 3.** (Continued)

| | Married %(CI) | Unmarried %(CI) |
|---|---|---|
| **Experienced major depressive disorder** | **13.6 (95% confidence limit 12.2-15.2)** | **5.1 (95% confidence limit 4.2-6.1)** |
| Discussed about how pregnancy occurs [b] | | |
| Yes | NA | 16.9 |
| No | NA | 4.9 |
| *Physical abuse by parents* | | |
| Physically hurt by parents in the last one-year [b] | | |
| Yes | NA | 10.6 |
| No | NA | 4.4 |
| *Reproductive history (only for married)* | | |
| Age at marriage (in years) [a] | | |
| 20-22 | 11.3 | NA |
| 18-19 | 12.0 | NA |
| 15-17 | 14.0 | NA |
| 7 to 14 | 16.6 | NA |
| Currently living with husband | | |
| Couple lives together | 13.4 | NA |
| Couple stays apart | 14.2 | NA |
| Reported last sexual intercourse [a] | | |
| Days ago | 12.7 | NA |
| Weeks ago | 15.1 | NA |
| Months ago | 13.7 | NA |
| Years ago | 19.8 | NA |
| Had experienced miscarriage/ stillbirth | | |
| Yes | 14.4 | NA |
| No | 13.4 | NA |
| Had experienced induced abortion [a] | | |
| Yes | 19.3 | NA |
| No | 13.4 | NA |
| Ever given birth to a live child | | |
| Yes | 13.4 | NA |
| No | 14.5 | NA |
| In-laws pressurized to conceive immediately or would call infecund [a] | | |
| Yes | 19.1 | NA |
| No | 12.3 | NA |

[a] significant association with experience of moderate to severe depression among young married women (Chi-square test, $p < 0.05$)

[b] Significant association with experience of moderate to severe depression among unmarried girls (Chi-square test, $p < 0.05$)

## Multilevel binary logistic regression

**Young married women.** Table 4 presents the results of multilevel binary logistic regression to understand the relationship between MDD and the explanatory variables for young married women. Model-0 or the 'Null model' is the intercept-only model (without any independent variable) that shows significant variability in MDD. The total computed variance of

**Table 4. Multilevel binary logistic regression analysis showing the association of having MDD with individual and community level variables among young married women.**

| Individual level variables | Model 0 | Model 1 | Model 2 | Model 3 |
|---|---|---|---|---|
| | | AOR (95% CI) | AOR (95% CI) | AOR (95% CI) |
| *Socio-economic characteristics* | | | | |
| Age in completed years | | | | |
| 18-23 | | 2.37 (1.18-4.78) * | | 2.32 (1.15-4.68) * |
| 12-17 | | Ref. | | Ref. |
| Social group | | | | |
| Scheduled castes (SC)/tribes (ST) | | 0.84 (0.63-1.12) | | 0.8 (0.56-1.14) |
| Other backward classes (OBCs) | | 0.69 (0.53-0.89) * | | 0.68 (0.53-0.88) * |
| General | | Ref. | | Ref. |
| Religion | | | | |
| Hindu | | 0.98 (0.76-1.27) | | 1.31 (0.96-1.79) |
| Non-Hindu | | Ref. | | Ref. |
| Completed years of schooling | | | | |
| Standard 1 to 8 | | 1.47 (1.14-1.89) * | | 1.48 (1.15-1.91) * |
| Standard 9 to 10 | | 1.63 (1.22-2.18) * | | 1.64 (1.22-2.19) * |
| Standard 11+ | | 1.57 (1.15-2.14) * | | 1.57 (1.15-2.14) * |
| No schooling | | Ref. | | Ref. |
| State specific wealth index | | | | |
| High | | 0.81 (0.65-1) | | 0.81 (0.65-1.01) |
| Middle | | Ref. | | Ref. |
| Low | | 0.9 (0.74-1.1) | | 0.91 (0.74-1.11) |
| Working status | | | | |
| Done any paid work in past one year | | | | |
| Yes | | 1.78 (1.1-2.88) * | | 1.78 (1.1-2.89) * |
| No | | Ref. | | Ref. |
| Currently looking for a job | | | | |
| Yes | | 2.07 (1.69-2.54) * | | 2.07 (1.69-2.54) * |
| No | | Ref. | | Ref. |
| *Social capital* | | | | |
| Currently attending school/ educational course | | | | |
| Yes | | 0.59 (0.39-0.89) * | | 0.59 (0.39-0.9) * |
| No | | Ref. | | Ref. |
| Number of friends | | | | |
| 0–3 | | Ref. | | Ref. |
| 4 or more | | 0.82 (0.69-0.96) * | | 0.81 (0.69-0.96) * |
| Been a member of youth/women's/girls' group in the last three years | | | | |
| Yes | | 1.14 (0.87-1.48) | | 1.15 (0.88-1.51) |
| No | | Ref. | | Ref. |
| *Agency* | | | | |
| Took decision on how money earned the last year should be used | | | | |
| Yes | | 0.7 (0.42-1.17) | | 0.7 (0.42-1.16) |
| Did not decide herself or did not earn any money | | Ref. | | Ref. |
| Took decision on how much education should have/till which standard | | | | |
| Yes | | 0.8 (0.66-0.96) * | | 0.79 (0.66-0.95) * |
| No | | Ref. | | Ref. |
| Took decision about who would be friends | | | | |

*(Continued)*

**Table 4.** (Continued)

| Individual level variables | Model 0 | Model 1 | Model 2 | Model 3 |
|---|---|---|---|---|
| | | AOR (95% CI) | AOR (95% CI) | AOR (95% CI) |
| Yes | | 0.91 (0.61-1.35) | | 0.93 (0.62-1.38) |
| No | | Ref. | | Ref. |
| Took decision about making major household purchases | | | | |
| Yes | | 1.14 (0.93-1.39) | | 1.14 (0.94-1.39) |
| No | | Ref. | | |
| Took decision about whether to work or stay at home | | | | |
| Yes | | 1.05 (0.86-1.28) | | 1.05 (0.86-1.29) |
| No | | Ref. | | Ref. |
| Took decision about seeking treatment if feeling sick | | | | |
| Yes | | 1.02 (0.83-1.25) | | 1.02 (0.83-1.25) |
| No | | Ref. | | Ref. |
| *Self-efficacy scores* | | | | |
| 5–15 | | Ref. | | Ref. |
| 16–20 | | 0.72 (0.6-0.86) * | | 0.72 (0.6-0.87) * |
| 21-24 | | 0.74 (0.6-0.93) * | | 0.75 (0.6-0.93) * |
| *Experience of harassment* | | | | |
| Experienced telephonic or online harassment (in last three years) | | | | |
| Yes | | 2.24 (1.68-2.98) * | | 2.28 (1.71-3.04) * |
| No | | Ref. | | Ref. |
| *Physical activity* | | | | |
| Duration of heavy work | | | | |
| 1-60 hours | | 1.49 (1.2-1.85) * | | 1.49 (1.2-1.85) * |
| No work (or <1 hour) | | Ref. | | Ref. |
| Duration of screen time | | | | |
| More than 15 hours | | 1.17 (0.86-1.6) | | 1.14 (0.83-1.56) |
| 3 to 15 hours | | 0.83 (0.7-0.99) * | | 0.83 (0.69-0.99) * |
| 0-3 hours | | Ref. | | Ref. |
| *Reproductive history* | | | | |
| Age at marriage | | | | |
| 7-14 year | | 1.48 (0.93-2.34) | | 1.48 (0.94-2.35) |
| 15-17 year | | 1.24 (0.83-1.86) | | 1.24 (0.82-1.86) |
| 18-19 year | | 1.09 (0.72-1.64) | | 1.08 (0.72-1.64) |
| 18 or more | | Ref. | | Ref. |
| Currently living with husband | | | | |
| Couple stays together | | 0.89 (0.73-1.07) | | 0.88 (0.73-1.07) |
| Couple lives apart | | Ref. | | Ref. |
| Reported last sexual intercourse | | | | |
| Days ago | | Ref. | | Ref. |
| Weeks ago | | 1.3 (1.04-1.61) * | | 1.29 (1.04-1.6) * |
| Months ago | | 1.11 (0.91-1.36) | | 1.1 (0.9-1.35) |
| Years ago | | 1.35 (0.8-2.3) | | 1.33 (0.78-2.26) |
| Had experienced miscarriage/ stillbirth | | | | |
| Yes | | 1.03 (0.85-1.25) | | 1.02 (0.84-1.25) |
| No | | Ref. | | Ref. |
| Had experienced induced abortion | | | | |
| Yes | | 1.31 (0.9-1.92) | | 1.29 (0.88-1.88) |

*(Continued)*

**Table 4.** (Continued)

| Individual level variables | Model 0 | Model 1 | Model 2 | Model 3 |
|---|---|---|---|---|
|  |  | AOR (95% CI) | AOR (95% CI) | AOR (95% CI) |
| No |  | Ref. |  | Ref. |
| Ever given birth to a live child |  |  |  |  |
| Yes |  | 0.93 (0.75-1.15) |  | 0.93 (0.75-1.15) |
| No |  | Ref. |  | Ref. |
| In-laws pressurized to conceive immediately or would call infecund |  |  |  |  |
| Yes |  | 1.69 (1.39-2.05) * |  | 1.68 (1.38-2.04) * |
| No |  | Ref. |  | Ref. |
| **Community level variables (aggregated at PSU level)** |  |  |  |  |
| Experienced any sexual assaults in last three years |  |  |  |  |
| less than 25% participants |  |  | Ref. | Ref. |
| 25–50% of participants |  |  | 0.77 (1.59-1.06) | 1.06 (0.73-1.55) |
| more than 50% of participants |  |  | 1.29 (3.66-2.15) * | 2.15 (1.24-3.73) * |
| Experienced dowry-related humiliation |  |  |  |  |
| less than 25% participants |  |  | Ref. | Ref. |
| 25–35% of participants |  |  | 1.24 (2.75-1.74) * | 1.74 (1.15-2.64) * |
| more than 35% of participants |  |  | 0.8 (1.88-1.1) | 1.1 (0.71-1.72) |
| Reported IPV in the last one year |  |  |  |  |
| less than 15% participants |  |  | Ref. | Ref. |
| 15-25% of participants |  |  | 0.72 (1.75-1.08) | 1.08 (0.68-1.73) |
| more than 25% of participants |  |  | 0.57 (1.43-0.88) | 0.88 (0.54-1.43) * |
| Boys with gendered attitude towards women |  |  |  |  |
| less than 25% of participants |  |  | Ref. | Ref. |
| 25 to 45% of participants |  |  | 0.64 (1.54-0.98) | 0.98 (0.62-1.56) |
| more than 45% of participants |  |  | 0.63 (1.56-1.01) | 1.01 (0.63-1.61) |
| **Model fit statistics*** |  |  |  |  |
| Variance (Intercept) | 0.602, <0.001* | 0.634, <0.001* | 0.524, <0.001* | 0.575, <0.001* |
| -2LL | 30124.056 | 31111.635 | 30224.511* | 31230.143 |
| AIC | 30126.057 | 31113.636 | 30226.512* | 31232.144 |
| ICC | 0.154 | 0.161* | 0.137 | 0.148 |
| PCV | Ref. | 0.053 | 0.129* | 0.044 |

The values with asterisks (*) indicate the best fit with respect to the corresponding indices.

-2LL: Compares the fit of two models by comparing their log likelihoods.

ICC: A measure of the proportion of variance that is explained by the group membership.

AIC: A measure to compare how well different statistical models fit the data.

PCV: A measure of the variability in the odds of the outcome variable at a given level after adding explanatory variables at that level with respect to the empty model.

15% in the outcome variables was reported by between-community variation of characteristics (ICC: 0.154). The following model (Model 1) includes all the individual-level variables as fixed factors. Model-fit statistics indicate that 16% of the variation in the level of major depression was accountable to the community-level differences among the young married women (ICC: 16.1). The PCV in this model shows that the individual-level characteristics of married participants explain around 5.3% of the variance. The ICC in Model 2 which comprised only the community-level aggregated variables exhibit that the differences between communities are responsible for almost 14% of the variation in the level of depression (ICC:13.7%).

The final model, i.e., Model 3, included all the individual and community-level variables together. The ICC value of these models could explain almost 15% of the variability in experiencing the MDD (ICC: 14.8). In the fixed effect analysis of the final model, young married women from OBC category had significantly lower odds of depression (AOR: 0.68, 95% CI 0.52–0.88). Age was a significant predictor with an odd of 2.32 (95% CI 1.15–4.67) for 18-23 years of the participants. Young married women with formal education up to primary, secondary, and above were 48.2%, 63.5%, and 56.7% more likely to be at higher levels of depression. Considering their working status, married women who had done any paid work in the previous year and those looking for a job had higher odds of experiencing major depression (AOR: 1.78, 95% CI 1.10–2.88 and AOR: 2.07, 95% CI 1.68–2.53). For the indicator of social capital, individuals currently enrolled in school or any other educational courses were 40.6% less likely to have MDD (AOR: 0.68, 95% CI 0.52–0.88). Likewise, having more than three friends significantly reduced the odds of MDD by 19.9% (AOR: 0.81, 95% CI 0.68–0.95). Young married women who could participate in decision-making related to their education had lower odds of suffering from MDD (AOR: 0.79, 95% CI 0.65–0.95). Likewise, those who had high self-efficacy scores (16–20 and 21–24) showed lower odds of experiencing the MDD of major depression (AOR: 0.72, 95% CI 0.59-0.86 and AOR: 0.74, 95% CI 0.59–0.93). Young married women who experienced any telephonic or online harassment in the last three years (preceding the date of the survey) were more likely to be at a higher level of depression by 2.28 times (AOR: 2.28, 95% CI 1.71–3.04). Young married women having 1–60 hours of heavy work were found to have higher odds (AOR: 1.48, 95% CI 1.19–1.84). On the contrary, those who watched the screen for 3 to 15 hours had lower odds of having MDD (AOR: 0.82, 95% CI 0.69–0.98).

Considering the reproductive history of the young married women, the frequency of inter-partner physical intimacy (weeks ago) remained a significant predictor of the level of depression (AOR:1.29, 95% CI 1.03–1.60). Those who had an abortion and received pressure from their in-laws to conceive or to be called infecund showed significantly higher odds of having major depressive depression behavior (AOR:1.28, 95% CI 0.87–1.88 and AOR: 1.67, 95% CI 1.38–2.03 respectively).

Amongst the community-level variables, the result shows that PSU-wise prevalence of IPV, dowry-related humiliation, and sexual assault (by a random person) were significant predictor of MDD among the young married women. Married women from the communities where the prevalence of sexual assaults was more than 50% had almost two times higher odds of having MDD. Similarly, those who are from communities where the prevalence of dowry-related humiliation ranges between 25% and 35%, were more likely to have MDD (AOR: 1.74, 95% CI 1.15–2.64). Our study estimated that there are approximately 27% of the studied PSUs where over 25% of the young married women reported IPV. In communities with the highest prevalence of IPV, odds of having MDD were significantly lower by 0.87 times (AOR: 0.87, 95% CI 0.54–1.42) The inclusion of one in the confidence interval of IPV is subjected to a P value of 0.053, which infer that the variable is near significant.

The AIC values were calculated for the consecutive models, and it depicts that Model 2, including only the community-level variables, had the smallest AIC. Therefore, this model is the best fit one, compared with other fitted models. Results of all the models present that both individual and community level variables have significant impacts on MDD in young married women.

**Unmarried girls.**    The random and fixed effect analysis of multilevel binary logistic regression of the unmarried girls was presented in Table 5. The null-model shows that without any independent variables, the odds of MDD significantly vary across the PSUs with a total computed variance of 22% (ICC: 0.22).

**Table 5. Multilevel binary logistic regression analysis showing the association of MDD with individual and community level variables among unmarried girls.**

| Individual level variables | Model 0 | Model 1 | Model 2 | Model 3 |
|---|---|---|---|---|
| | | AOR (95% CI) | AOR (95% CI) | AOR (95% CI) |
| *Socio-economic characteristics* | | | | |
| Age in completed years | | | | |
| 18-23 | | 1.66 (1.22-2.27) * | | 1.64 (1.2-2.24) * |
| 12-17 | | Ref. | | Ref. |
| Social group | | | | |
| Scheduled castes (SC)/tribes (ST) | | 1.09 (0.73-1.63) | | 0.55 (1.13-0.76) |
| Other backward classes (OBCs) | | 1.33 (0.96-1.85) | | 0.06 (1.37-0.99) |
| General | | Ref. | | Ref. |
| Religion | | | | |
| Hindu | | 0.77 (0.56-1.06) | | 0.19 (0.81-0.58) |
| Non-Hindu | | Ref. | | Ref. |
| Completed years of schooling | | | | |
| Standard 1 to 8 | | 0.59 (0.34-1.03) | | 0.63 (0.36-1.09) |
| Standard 9 to 10 | | 1.07 (0.59-1.94) | | 1.14 (0.62-2.06) |
| Standard 11+ | | 0.98 (0.52-1.84) | | 1.06 (0.56-2.01) |
| No schooling | | Ref. | | Ref. |
| State specific wealth index | | | | |
| High | | 0.72 (0.53-0.98) * | | 0.7 (0.52-0.95) * |
| Middle | | Ref. | | Ref. |
| Low | | 0.85 (0.63-1.15) | | 0.87 (0.64-1.18) |
| Working status | | | | |
| Done any paid work in past one year | | | | |
| Yes | | 1.16 (0.73-1.83) | | 1.15 (0.73-1.82) |
| No | | Ref. | | Ref. |
| Currently looking for a job | | | | |
| Yes | | 2.46 (1.79-3.36) * | | 2.42 (1.76-3.32) * |
| No | | Ref. | | Ref. |
| *Social capital* | | | | |
| Currently attending any school/ educational course | | | | |
| Yes | | 0.41 (0.22-0.79) * | | 0.41 (0.22-0.78) * |
| No | | Ref. | | Ref. |
| Number of friends | | | | |
| 4 or more | | 1.37 (1.07-1.74) * | | 1.37 (1.07-1.74) * |
| 0 to 3 | | Ref. | | Ref. |
| Been a member of youth/women's/girls' group in the last three years | | | | |
| Yes | | 1.61 (0.88-2.96) | | 1.62 (0.88-2.98) |
| No | | Ref. | | Ref. |
| *Agency* | | | | |
| Took decision on how money earned the last year should be used | | | | |
| Yes | | 0.77 (0.47-1.26) | | 0.75 (0.46-1.24) |
| Did not decide herself or did not earn any money | | Ref. | | Ref. |
| Took decision on how much education should have/till which standard | | | | |
| Yes | | 1.43 (1.07-1.92) * | | 1.43 (1.06-1.91) * |
| No | | Ref. | | Ref. |
| Took decision about who would be friends | | | | |
| Yes | | 0.79 (0.45-1.41) | | 0.8 (0.45-1.42) |

(*Continued*)

**Table 5.** (*Continued*)

| Individual level variables | Model 0 | Model 1 | Model 2 | Model 3 |
|---|---|---|---|---|
| | | AOR (95% CI) | AOR (95% CI) | AOR (95% CI) |
| No | | Ref. | | Ref. |
| Took decision about making major household purchases | | | | |
| Yes | | 0.96 (0.72-1.29) | | 0.96 (0.72-1.28) |
| No | | Ref. | | Ref. |
| Took decision about whether to work or stay at home | | | | |
| Yes | | 1.12 (0.84-1.49) | | 1.12 (0.84-1.5) |
| No | | Ref. | | Ref. |
| Took decision about seeking treatment if feeling sick | | | | |
| Yes | | 1.54 (1.16-2.05) * | | 1.55 (1.17-2.07) * |
| No | | Ref. | | Ref. |
| Self-efficacy scores | | | | |
| 5–15 | | | | |
| 16–20 | | 0.35 (0.26-0.47) * | | 0.35 (0.26-0.47) * |
| 21-24 | | 0.29 (0.2-0.4) * | | 0.28 (0.2-0.39) * |
| *Experience of harassment* | | Ref. | | Ref. |
| Experienced telephonic or online harassment (in last three years) | | | | |
| Yes | | 2.14 (1.46-3.13) * | | 2.2 (1.5-3.24) * |
| No | | Ref. | | Ref. |
| *Physical activity* | | | | |
| Duration of heavy work | | | | |
| 1-60 hours | | 1.83 (1.4-2.39) * | | 1.91 (1.47-2.5) * |
| No work (or <1 hours) | | Ref. | | Ref. |
| Duration of screen time | | | | |
| More than 15 hours | | 0.47 (0.3-0.73) * | | 0.44 (0.28-0.69) * |
| 3 to 15 hours | | 0.56 (0.43-0.73) * | | 0.54 (0.42-0.7) * |
| 0-3 hours | | Ref. | | Ref. |
| *Interaction with parents in last year* | | | | |
| Discussed about school performance | | | | |
| Yes | | 1.28 (0.69-2.38) | | 1.26 (0.68-2.33) |
| No | | Ref. | | |
| Discussed about friendship | | | | |
| Yes | | 2.17 (1.63-2.89) * | | 2.13 (1.6-2.84) * |
| No | | Ref. | | Ref. |
| Discussed about being teased | | | | |
| Yes | | 1.98 (1.5-2.6) * | | 1.96 (1.49-2.59) * |
| No | | Ref. | | Ref. |
| Past one year, discussed about menstruation | | | | |
| Yes | | 0.82 (0.64-1.06) | | 0.8 (0.62-1.04) |
| No | | Ref. | | Ref. |
| Discussed about how pregnancy occurs | | | | |
| Yes | | 2.21 (1.24-3.97) * | | 2.2 (1.23-3.97) * |
| No | | Ref. | | Ref. |
| *Physical abuse by parents* | | | | |
| Physically hurt by father or mother of both in the last one year | | | | |
| Yes | | 3.457 (2.55-4.69) * | | 3.49 (2.57-4.75) * |
| No | | Ref. | | Ref. |

(*Continued*)

**Table 5.** (Continued)

| Individual level variables | Model 0 | Model 1 | Model 2 | Model 3 |
|---|---|---|---|---|
| | | AOR (95% CI) | AOR (95% CI) | AOR (95% CI) |
| **Community level variable (aggregated at PSU level)** | | | | |
| Reporting of family violence: father beaten mother | | | | |
| Does not have parents or reported no violence | | | Ref. | Ref. |
| less than 10% of participants | | | 0.55 (0.31-0.98) * | 0.45 (0.24-0.85) * |
| 10% or more participants | | | 0.52 (0.29-0.94) * | 0.35 (0.19-0.68) * |
| Reporting of any kind of sibling inequality | | | | |
| less than 0-15% participants | | | Ref. | Ref. |
| 16–30% of participants | | | 0.98 (0.61-1.59) | 0.9 (0.53-1.51) |
| more than 30% of participants | | | 1.25 (0.74-2.13) | 1.11 (0.63-1.97) |
| Experienced any sexual assaults in last three years | | | | |
| Less than 25% participants | | | Ref. | Ref. |
| 25–50% of participants | | | 0.65 (0.41-1.04) | 0.65 (0.39-1.07) |
| more than 50% of participants | | | 1.02 (0.58-1.77) | 1.06 (0.58-1.95) |
| Experienced dowry-related humiliation | | | | |
| less than 25% participants | | | Ref. | Ref. |
| 25–35% of participants | | | 1.28 (0.79-2.08) | 1.04 (0.61-1.76) |
| more than 35% of participants | | | 1.62 (0.99-2.67) | 1.47 (0.86-2.52) |
| Reported IPV in the last one year | | | | |
| less than 15% participants | | | Ref. | Ref. |
| 15-25% of participants | | | 0.66 (0.41-1.08) | 0.69 (0.41-1.18) |
| more than 25% of participants | | | 0.41 (0.24-0.7) * | 0.42 (0.23-0.74) * |
| Boys with gendered attitude towards women | | | | |
| less than 25% of participants | | | Ref. | Ref. |
| 25 to 45% of participants | | | 1.02 (0.63-1.65) | 1.13 (0.67-1.91) |
| more than 45% of participants | | | 0.99 (0.58-1.7) | 1.04 (0.57-1.87) |
| **Model fit statistics*** | | | | |
| Variance (Intercept) | 0.928, <0.001* | 1.111, <0.001* | 0.839, <0.001* | 0.977, <0.001* |
| -2LL | 49188.240 | 54625.420 | 49660.286* | 55544.744 |
| AIC | 49190.240 | 54627.420 | 49662.286* | 55546.744 |
| ICC | 0.220 | 0.252* | 0.203 | 0.228 |
| PCV | . Ref. | -0.197 | 0.095* | -0.052 |

The values with asterisks (*) indicate the best fit with respect to the corresponding indices.

-2LL: Compares the fit of two models by comparing their log likelihoods.

ICC: A measure of the proportion of variance that is explained by the group membership.

AIC: A measure to compare how well different statistical models fit the data.

PCV: A measure of the variability in the odds of the outcome variable at a given level after adding explanatory variables at that level with respect to the empty model.

In the following model, which includes all individual-level variables, the model-fit statistics, around 25% (ICC: 0.252) of the variation was attributable to community-level differences among the unmarried girls. The following model (Model 2) includes only community-level variables, showing the association of prevalence of family violence, dowry-related humiliation, and IPV. This model also showed that 20.3% of the total variation in major depression was due to community-level factors.

The third and final model, where both individual and community-level variables were included, showed 22.8% of the total variance in MDD could be attributable to the community

level [62]. The fixed effect analysis reveals that unmarried girls belonging to the higher category of wealth index had lower odds (AOR: 0.05, 95% CI 1.37–0.99 and AOR: 0.69, 95% CI 0.51–0.94) of MDD. Age was found as a significant predictor since unmarried girls aged 18–23 years were 63% more likely to have MDD (AOR: 1.63, 95% CI 1.19–2.24). Those who were looking for a job showed 2.4 times (AOR: 2.41, 95% CI 1.76–3.31) higher odds of MDD. When social capital was considered, currently enrolled students (in school or any educational course) had around 60% less odds of having MDD (AOR:0.41, 95% CI 0.21–0.78). Contrastingly, those who have more than three friends had 1.37 times higher odds of MDD (AOR: 1.36, 95% CI 1.07–1.74). The attributes of agency such as decision-making regarding the level of education and health seeking showed lower odds of suffering from major depression (AOR: 1.42, 95% CI 1.06–1.91 and AOR: 1.55, 95% CI 1.16–2.06). Likewise, unmarried girls with higher self-efficacy scores (16–20 and 21–24) had lower odds of having MDD by 0.35 (95% CI 0.26–0.47) and 0.27 (95% CI 0.19–0.39) times. In terms of violence and harassment, those encountered telephonic or online harassment and physical abuse (by parents) had more than two times and three times higher odds of MDD (AOR: 2.20, 95% CI 1.49–3.23 and AOR: 3.49, 95% CI 2.57–4.74). Considering the parameters of physical activity, hours of heavy physical work increased the odds of depression by 91.4% (AOR: 1.91, 95% CI 1.46–2.50), whereas more screen time (3-15 hour. and more than 15 hour) could lower the odds by 45.8% and 56.2% (AOR: 0.54, 95% CI 0.41–0.70 and AOR: 0.43, 95% CI 0.28–0.68). Unmarried girls who interacted with their parents regarding friendship and how pregnancy occurs showed higher odds of MDD of major depression (AOR: 2.13, 95% CI 1.59-2.84 and AOR:2.20, 95% CI 1.22–3.96). The odds of having MDD were significantly higher among the unmarried girls living in those PSUs that had a higher percentage of family violence (less than 10% and 10% or more), compared to the PSUs that reported no family violence (AOR: 0.45, 95% CI, 0.24–0.85 and AOR: 0.35, 95% CI 0.18–0.67). Participants from the communities with IPV more than 25% of prevalence, had significantly lower odds of MDD (AOR: 0.41, 95% CI 0.23–0.74).

## Discussion

Following the EST model the study revealed how both individual and community level variables explained the determinants of MDD for young married women and unmarried girls. At the micro-system, the individual centric factors such as age, participation in educational decision making, social capital (like currently attending school/educational course and number of friends), self-efficacy, telephonic harassment, and physical activity were associated with MDD. Wealth index, job seeking, participation in health seeking decision making, parental interactions and physical abuse (for unmarried girls) and education, time of last sexual intercourse, pressure from in-laws to conceive (for young married women) were associated with MDD. At the exo-system, in case of young married women, the community-level variables reflect a regressive social environment such as higher prevalence of dowry-related humiliation, IPV, and sexual assaults, and are significantly associated with MDD. For the unmarried girls, family violence (father beaten mother) is the significant predictor of MDD.

The random effect of both the MLM delineated the significance of both individual and community level variables in explaining the susceptibility to MDD. Variables that operate at the exo-system or community-level, such as young married women who reside in communities that are reported to have high rates of sexual assaults and dowry-related humiliation (25%-35%), the odds of having MDD is higher. This finding reflects that how degraded values and regressive social environment make young married women prone to MDD [54, 55, 79]. Participants of both the groups, participants representing communities that reported >25% IPV and for the unmarried girls, representing communities where male perpetuated family violence

(father beaten mother) have been reported, showed lower odds of experiencing MDD. Studies show that domestic violence in terms of IPV or vehemence between parents contributed to individuals' psychological distress [80, 81], however, reverse trend was observed in this study, owing to community realm. The ability to report domestic violence reflects a strong trajectory of women empowerment imbibed within the social environment [82–84]. Earlier studies have shown that the embodied resources (like self-care, agency, self-efficacy) of empowerment effectively contribute to encountering stress and depression [85, 86]. Our study corroborates with this trend. Association between MDD and women empowerment is reaffirmed by certain individual-centric factors. At the micro-level, variables that reflect women's empowerment (eg. human agency, social capital) showed significant association with MDD. More research is required to understand how these variables nested at the individual level of ecological system reforming the higher level of social ecology.

An increased odds of MDD among the adults (≥18 years) could be attributed to their socio-biological changes at their post-pubertal phase, characterized by more social under-standing, self-awareness, and modification in encephalon related to responses in increased stress levels [87–89]. Similar association has been reported from earlier studies where older female adolescents have lower self-compassion, life satisfaction, and higher emotional stress [90–93]. Our findings reveal higher odds of MDD among participants with higher educational level and who are in paid job or are in search of jobs, unlike some previous findings [94, 95]. Thus, in the present study, increase in the level of formal education increases the odds of having MDD which could be attributed to perceived educational stipulation and ambition, coupled with an insecure or unsatisfactory job [96–98]. Similarly, a plateau or worsening of mental health can also result from unemployment and the stress related to job search [99, 100]. Conforming to some earlier studies, this study revealed the multifaceted burden of socio-economic and psychological stress resulting from "unemployment scarring" [101–103].

Relatively higher wealth status is likely to be associated with better mental health outcomes by provoking more comprehensive access to life opportunities and resources for easier adaptation in challenging socio-political circumstances [104–106]. The findings of this study corroborate to the previous works that reported a lower risk of depression among unmarried girls belonging to higher socioeconomic status [107–109]. Among the young married group, participants that represent the 'other backward community' (OBC) have significantly lower odds of MDD. This contradicts to the prior understanding that affiliation to lower social strata and ignorance could potentially be a stress factor [110, 111]. Studies suggest links between minority status and higher rates of psychological stress subjected to factors such as intercultural conflict, neighborhood discrimination, and unfair treatment [112]. However, the people of OBC category are numerically dominant in the states of Bihar and Uttar Pradesh (RGI 2011), which might have provided a preventive effect on psychological stress through fostering a sense of social cohesion [113].

In the era of profuse exposure to virtual spaces and remote communication, negative inter-actions or cyberbullying is mediating a wide range of intrusions causing emotional or psycho-social stress and social anxiety. Our study confirms similar findings where cyberbullying or telephonic harassment has an intimidating effect to disrupt the personal mental space and causing MDD [114–116], irrespective of the marital status of the studied participants.

Our findings reveal an unusual trend between engagement in heavy physical activity and MDD among both the groups, contradicting to most of the earlier works that establish an endocrinal benefit of physical exercise in reducing anxiety and depression [117, 118]. For young married women, heavy physical activity often refers to their household workload, instead of sports or exercise. The burden and fatigue of household workload harness poorer psychological health [119]. In addition, getting married at a very young age and consecutive

engagement in domestic tasks inherently causes emotional exhaustion [120]. The bivariate analyses revealed that the prevalence of MDD was the highest among the young married women who got married between the age of 7 and 14 years, but reduces with successive increase in age at marriage. A similar trend was also found in the association between higher duration of physical exercise and MDD among unmarried girls. The likelihood of MDD was higher for the unmarried girls who are doing physical exercise. This could be subjected to weariness and exhaustion of "overtraining syndrome" [121]. Likewise, the hours of screentime, in the form of watching movies or having thought-provoking experiences for both married and unmarried participants, acts as a source of amusement or recreation [122–124]. Unlike studies that showed increased odds of depression with higher screentime owing to sedentary leisure pursuits, insomnia, and social isolation [125, 126], our findings suggest that screentime (within 3-15 hours) has some protective effect on MDD, subjected to connectivity, entertainment, and relaxation [127]. Therefore, the content of screen time, rather than simply the duration, may be a more nuanced factor influencing mental health.

The attributes of social capital facilitate resources linked to the durable social network and enable coercion in functioning coordinated action [128]. This study affirms that being enrolled in an institutionalized education system and having a higher number of friends (more than three), which represent individuals' structural social capital [129] showed a lower risk of experiencing MDD irrespective of the marital status of the studied participants. A recent study [130] showed that social capital in terms of association with such institutionalized structures brings about an effective intervention for improved mental health outcomes. Contrary to the earlier works that showed association between parental interaction and MDD, we found that unmarried girls who interacted with parents about themselves being teased and their friendship issues, and discussed about how pregnancy occurs showed twice as a higher odd of MDD. Adolescence is characterized by a gradual decline of parental influence on individuals' life and optimizing the thirst for personal autonomy and empowerment [131]. Previous research reveals that if parental interaction lacks emotional warmth or becomes strict in communication it can critically impact adolescents' mental health [132]. Since the UDAYA survey data have not measured the quality of parental interaction, we are not in a position to provide a more detailed explanation behind our findings. Incidentally, girls who are hurt or abused by parents were more likely at a higher risk of MDD. Therefore despite having parental interaction, disapproving parenting along with domination, over-criticizing, instead of motivation or appraisal worsens the expectation of tangible parental support from the social network impeding the intimate and immediate world [133, 134]. Our findings also reflect lower odds of moderate to severe depression among married and unmarried participants who had higher self-efficacy and who could participate in education and health seeking (for the unmarried girls only) related decision-making. This is in line with the earlier works that proclaim that the command over human agency and autonomy act as a protective buffer in the adaptive course of psychosocial well-being [82, 135, 136].

Among the young married women who reported to have sexual intercourse two to three times a month were more likely at a risk of MDD. It could be likely that higher sexual satisfaction in terms of regularity might have beneficial effects on psychological well-being [136–138]. We further found that young married women who were pressurized to conceive immediately after marriage had higher odds of having MDD. In most South Asian societies, the prevailing social norm is to start childbearing immediately after marriage [139, 140]. Deviation from these translates into egoistic feelings of becoming a socially outcast [141]. Consequently, vilification from family and elders becomes stressful for young married women at household and social levels.

The National Mental Health Programme (NMHP) in 1982 marked as an inception of the initiatives to address mental health issues in India. Under NMHP, the District Mental Health Programme provides community-based services, upgrading mental health facilities and quality of care across primary, secondary, and tertiary levels [142, 143]. The National Mental Health Policy (formulated in 2014) and the Mental Healthcare Act (launched in 2017) introduced a rights-based approach, aiming to reduce stigma, regulate better mental health care, and improve services [143, 144]. Additionally, adapting to changing communication landscapes, the National Tele Mental Health Programme was launched in 2020 in response to COVID-19, offering wider access through online counseling services [145–147]. Several non-governmental organizations like National Institute of Mental Health and Neurosciences, or non-profit organization such as The Live Love Laugh Foundation, Sangath, or Anjali, focus on adolescent and young adult mental health, offering awareness, specialized programs, evidence-based interventions, and holistic care.

Despite such efforts, according to the National Mental Health Survey 2015–2016, approximately 7.3% of adolescents and 10.6% of young adults suffer with mental health issues. The prevalence rates in urban metropolitan areas are twice as high [13, 148]. This crisis worsens with a very high treatment gap, poor evidence-based treatment, and gender differentials [149, 150]. Prevention and early intervention of depression among young individuals would be possible through understanding the predisposing factors and early symptoms of depression. Our findings suggest, that addressing domain-specific determinants of depression, pertaining to the economy, education, employment, and empowerment alongside fostering the environment in larger social context such as schools, families, and communities, is essential. India has severe dearth of skilled mental health professionals and services [151]. Therefore, training could be provided to the non-specialist, such as community health workers using digital technologies [152] to build awareness and remove stigma at the grassroot level, in admitting and seeking care for mental health issues [153, 154]. The school-based and community-based social interventions have been showing promising results in this regard.

The application of EST model allowed us to examine how each ecological level of social environment explained the experience of MDD among both the groups of study participants, Findings could be considered under certain limitations. The PHQ-9 score used to detect MDD is based on self-reported responses about the frequency of experiencing MDD instead of any psychometric test that offers a clinical diagnosis. Few explanatory variables were collected within a temporal frame of three years (or one year), whereas, MDD was estimated based on the experience in the last 15 days. Such disparity might withhold the causal inference. Owing to the study design of UDAYA, the authors could not control the effects of pre-existing medical conditions such as genetic predisposition or physical disability. Further, the retrospective responses might face the animosity of recall bias. Certain explanatory variables such as social capital and human agency were composed using the available variables such as number of friends, association with youth groups, participation in decision-making, self-efficacy, etc. are the very limited aspects of the concerned domains.

## Conclusion

The study revealed that the prevalence of MDD stems from the dynamic and active engagement with individual and community-level environments. India has embraced a comprehensive approach to tackling mental health issues, with various initiatives spanning several ministries, which presents significant opportunity to establish a transformative youth mental health care system within the current infrastructure. However, reimagining the approach

requires collective efforts at various levels of social ecology for the betterment of mental health profile of adolescent and young adult individuals.

## Supporting information

**S1 Table. Variables used to compute level of depression using the PHQ-9 scoring method.** (DOCX)

**S2 Table. List of variables and assessment of composite score of Self-efficacy.** (DOCX)

**S3 Table. List of variables, assessments, and categories used to compute IPV against the young married women.** (DOCX)

**S4 Table. Variables and assessments for evaluating men's opinion about women in work and family life.** (DOCX)

**S5 Table. Variables that were used to assess perceived sibling inequality by the participants.** (DOCX)

**S1 File. List of independent variables (Individual level).** (DOCX)

## Acknowledgments

We would like to express our sincere gratitude to Population Council for granting us access to the data from the 'Understanding the Lives of Adolescents and Young Adults (UDAYA) in Bihar and Uttar Pradesh' study, which served as the primary source for this research. Additionally, we extend our heartfelt thanks to Dr. Priya Nanda for her valuable review of the manuscript and providing critical suggestions.

## Author Contributions

**Conceptualization:** Joyeeta Thakur, Neelanjana Pandey, Arupendra Mozumdar.

**Data curation:** Shromona Dhara.

**Formal analysis:** Shromona Dhara.

**Methodology:** Joyeeta Thakur, Neelanjana Pandey, Arupendra Mozumdar.

**Supervision:** Neelanjana Pandey, Arupendra Mozumdar, Subho Roy.

**Writing – original draft:** Shromona Dhara, Joyeeta Thakur.

**Writing – review & editing:** Arupendra Mozumdar, Subho Roy.

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
