## [Decision Letter · Decision Letter 0]

16 Feb 2024

PONE-D-23-28571

Prevalence of Major Depressive Symptoms and its Determinants among Young Married Women and Unmarried Girls: Findings from the Second Round of UDAYA Survey

PLOS ONE

Dear Dr. Dhara,

Thank you for submitting your manuscript to PLOS ONE. After careful consideration, we feel that it has merit but does not fully meet PLOS ONE’s publication criteria as it currently stands. Therefore, we invite you to submit a revised version of the manuscript that addresses the points raised during the review process.

We look forward to receiving your revised manuscript.

Kind regards,

Pradeep Kumar, Ph.D.

Academic Editor

PLOS ONE

3. For studies involving third-party data, we encourage authors to share any data specific to their analyses that they can legally distribute. PLOS recognizes, however, that authors may be using third-party data they do not have the rights to share. When third-party data cannot be publicly shared, authors must provide all information necessary for interested researchers to apply to gain access to the data. (https://journals.plos.org/plosone/s/data-availability#loc-acceptable-data-access-restrictions) 

Reviewers' comments:

**Comments to the Author**

1. Is the manuscript technically sound, and do the data support the conclusions?

Reviewer #1: Yes

Reviewer #2: Partly

2. Has the statistical analysis been performed appropriately and rigorously? 

Reviewer #1: Yes

Reviewer #2: Yes

3. Have the authors made all data underlying the findings in their manuscript fully available?

Reviewer #1: Yes

Reviewer #2: Yes

4. Is the manuscript presented in an intelligible fashion and written in standard English?

Reviewer #1: Yes

Reviewer #2: No

5. Review Comments to the Author

Reviewer #1: 1. The title of the article includes the phrase ‘major depressive symptoms’ but I do not find it defined anywhere in the text. This phrase is quite confusing as it matches only partially with the standard International Classification of Disease (ICD) terminology. ICD uses the phrase major depressive disorders. Since authors have used PHQ-9, I would suggest them to stick with the word ‘depression’ or the phrase ‘major depressive disorder’, but not ‘major depressive symptom’.

2. Abstract’s methodology section does not contain any information on what your dependent variable is and how it has been defined or created.

3. Abstract: Please include the age group these women belong to. Also, the year UDAYA data is from.

4. Abstract: ‘major depression’ does not make sense. Please choose one word or phrase, define it properly, and use it throughout the text to avoid confusing the reader.

5. Please include only major findings and do provide statistical details, for example, odds ratio and their p-values or confidence intervals, in the abstract.

6. targeted interventions – who needs to be targeted as per your findings? You may restructure the conclusive sentences to make it clear.

7. Approximately, 13.6% and 5.1% of young married 38 women and unmarried girls, respectively, experienced major depressive symptoms. The word approximately does not seem appropriate here as 13.6% is quite a clear number. It is not approximate.

8. Please include confidence intervals for both of these figures – 13.6 and 5.1.

9. Line 57 – avoid beginning your sentences with BUT

10. Introduction is weak and not structured well. Need to be reworked.

11. A lot of previous work on prevalence of major depressive disorder/depression has been ignored in the introduction. Please search thoroughly and cite them in the introduction.

12. After national mental health survey 2015-16, a major study by India State Level Disease Burden Initiative was published in 2020 where state wise data on 9 mental disorders, including major depressive disorders has been is presented. This was published in Lancet Psychiatry. Here is the link: https://www.thelancet.com/journals/lanpsy/article/PIIS2215-0366(19)30475-4/fulltext

13. Theoretical framework does not need to have a heading of its own. Let it flow within the introduction. Same with hypothesis heading.

14. The multicollinearity among the independent variables was 210 measured through the variance inflation factor (VIF > 10) and its respective tolerance (<0.1). Why have you included “(VIF > 10)” in your sentence?

15. Please describe the nested structure and provide figures (numbers) for each level.

16. Line 211 Explain what is mixed-effects? Is your model a random intercept or random slope model?

17. Lines - more IPV and less IPV categories? The use of more or less does not seem appropriate.

18. Line – 179: created by merging three questions – use appropriate language. You don’t merge questions to create a new variable.

19. Line 242: pay attention to grammar and tenses.

20. Line 244: ICC and PCV: How are they different from each other?

21. Odds ratio should be up to 2 digits after decimal. Three is too many.

22. Line 441: less odds should smaller odds. Restructure this sentence “Respondents from the communities with more than 25% prevalence of IPV, were significantly less odds of having symptoms of major depression” – does not make sense in its current form.

23. Clarify the meaning of 'social ecology' in line 448 to enhance reader understanding.

24. Line 416: “The third and final model, where both individual and community-level variables were included, showed 22.3% of the unexplained variation in depression could be attributable to community and household-level factors together.” Should it be “unexplained variation” or explained variation?

25. Table 3b: Model fit statistics – what is (Int.)?

26. Provide explanations for ICC, LL, AIC, BIC, etc., at the bottom of the table to enhance reader comprehension.

27. On what basis have you selected community level variables? Please include this in the manuscript.

28. Discussion should also include a paragraph or two focusing on the policy implications of your findings. Frame this in the context of ongoing governmental and non-governmental efforts in the country.

29. Line 555: “show a different set of predictors and their associations” – does not make sense – please rewrite this sentence. What do you mean by “different set” – different from???

30. Line 598: “The ongoing program on mental health issues at state level should focus on these findings and implement their intervention so that we can properly address this public health concern.” – program should focus on …findings??? It is unclear as to what authors want to convey by this sentence. Please rewrite this sentence. I would suggest you re-write the conclusion.

31. Why did authors apply multilevel models? The rationale behind choosing this kind of model must be provided in the text.

Reviewer #2: The introduction and results sections of your manuscript are well written. However, the discussion section lacks sufficient linkage to the existing literature, resulting in a gap in contextual grounding.

Create coherence in the discussion, identify the findings that conflict with other findings, and sufficiently discuss the reason for the incongruence.

Line 444-447: Summarise your key findings

Line 448: what do you mean by higher level.

Line 448: Replace the word “enlightened” with highlighted/delineated.

Line 451: cite the evidence of "degraded values and regressive community" from your study or else this is blanket statement.

Line 454: are you reporting lower depression in the instance of male perpetuated violence against women in the family??

Line 453-455: ?

Line 478: Does it prevent mental health ?

Line 480-489: delineate the relevant findings , add context and implication of findings with respect to variables related to education.

Line 501-511: incoherent sentences.

Line 510: if so, contextualise with studies that suggest association between increase in screen time and mental health.

6. PLOS authors have the option to publish the peer review history of their article (what does this mean?). If published, this will include your full peer review and any attached files.

Reviewer #1: **Yes: **Aditya Singh

Reviewer #2: No

---

## [Author Response · Author response to Decision Letter 0]

24 Apr 2024

We express our sincere gratitude to the reviewers for their insightful comments. We have closely gone through all the comments and revised the manuscript accordingly. In the docx file named 'Response to Reviewers" we have stated the responses to each point raised by the reviewers.

---

## [Decision Letter · Decision Letter 1]

12 Jun 2024

Prevalence of Major Depressive Disorder and its Determinants among Young Married Women and Unmarried Girls: Findings from the Second Round of UDAYA Survey

PONE-D-23-28571R1

Dear Dr. Dhara,

We’re pleased to inform you that your manuscript has been judged scientifically suitable for publication and will be formally accepted for publication once it meets all outstanding technical requirements.

Kind regards,

Pradeep Kumar, Ph.D.

Academic Editor

PLOS ONE

Reviewers' comments:

Reviewer's Responses to Questions

**Comments to the Author**

1. If the authors have adequately addressed your comments raised in a previous round of review and you feel that this manuscript is now acceptable for publication, you may indicate that here to bypass the “Comments to the Author” section, enter your conflict of interest statement in the “Confidential to Editor” section, and submit your "Accept" recommendation.

Reviewer #1: All comments have been addressed

Reviewer #2: All comments have been addressed

2. Is the manuscript technically sound, and do the data support the conclusions?

Reviewer #1: Yes

Reviewer #2: Yes

3. Has the statistical analysis been performed appropriately and rigorously? 

Reviewer #1: Yes

Reviewer #2: Yes

4. Have the authors made all data underlying the findings in their manuscript fully available?

Reviewer #1: Yes

Reviewer #2: Yes

5. Is the manuscript presented in an intelligible fashion and written in standard English?

Reviewer #1: Yes

Reviewer #2: Yes

6. Review Comments to the Author

Reviewer #1: All the comments have been addressed to the satisfaction of the reviewer. Authors have incorporated all the suggestion as well.

Reviewer #2: (No Response)

7. PLOS authors have the option to publish the peer review history of their article (what does this mean?). If published, this will include your full peer review and any attached files.

Reviewer #1: **Yes: **Aditya Singh

Reviewer #2: No

---

## [Editor Report · Acceptance letter]

23 Jun 2024

PONE-D-23-28571R1 

PLOS ONE

Dear Dr. Dhara, 

I'm pleased to inform you that your manuscript has been deemed suitable for publication in PLOS ONE. Congratulations! Your manuscript is now being handed over to our production team.

Kind regards, 

on behalf of

Dr. Pradeep Kumar 

Academic Editor

PLOS ONE